Chronic playback of boat noise does not impact hatching success or post-hatching larval growth and survival in a cichlid fish

Bruintjes Rick 1 2 rbruintjes@yahoo.com
Radford Andrew N. 1
1 School of Biological Sciences, University of Bristol , Bristol , United Kingdom
2 Biosciences, College of Life and Environmental Sciences, University of Exeter , Exeter , United Kingdom
Whitehead Andrew
Electronic publication date: 2014 Sep 25
Publication date: 2014
Volume: 2
Electronic Location ID: e594
Received 2014 Jul 3; Accepted 2014 Sep 3
Copyright: © 2014 Bruintjes and Radford
Copyright year: 2014
Copyright holder: Bruintjes and Radford
License: This is an open access article distributed under the terms of the Creative Commons Attribution License, which permits unrestricted use, distribution, reproduction and adaptation in any medium and for any purpose provided that it is properly attributed. For attribution, the original author(s), title, publication source (PeerJ) and either DOI or URL of the article must be cited.
License URL: https://creativecommons.org/licenses/by/4.0/

Keywords: Cichlidae, Anthropogenic noise, Growth, Development, Offspring survival, Long-term noise exposure, Sound, Teleost, Fitness, Vertebrates, Lake Tanganyika

Funding: SNF PBBEP3_134918 PADI Aware 5099 ASAB Research Grant Funding was provided by SNF (PBBEP3_134918), PADI Aware (5099) and an ASAB Research Grant. The funders had no role in study design, data collection and analysis, decision to publish, or preparation of the manuscript.

==============================
Anthropogenic (man-made) noise has been shown to have a negative impact on the behaviour and physiology of a range of terrestrial and aquatic animals. However, direct assessments of fitness consequences are rare. Here we examine the effect of additional noise on early life stages in the model cichlid fish, Neolamprologus pulcher. Many fishes use and produce sounds, they are crucial elements of aquatic ecosystems, and there is mounting evidence that they are vulnerable to anthropogenic noise; adult N. pulcher have recently been shown to change key behaviours during playback of motor boat noise. Using a split-brood design to eliminate potential genetic effects, we exposed half of the eggs and fry from each clutch to four weeks of playbacks of noise originally recorded from small motor boats with the other half acting as a control (receiving no noise playback). There was no significant effect of additional noise on hatching success or fry survival, length or weight at the end of the exposure period. Although care should be taken not to generalize these findings on a single species from a laboratory study, our data suggest that moderate noise increases do not necessarily have direct negative impacts on early-life survival and growth. Further studies on a range of species in natural conditions are urgently needed to inform conservation efforts and policy decisions about the consequences of anthropogenic noise.

Introduction

Noise-generating human activities, such as transportation, urbanisation and resource exploitation, have altered the acoustic environment in many terrestrial and aquatic environments around the globe (Normandeau Associates Inc., 2012; Watts et al., 2007). Consequently, anthropogenic or man-made noise is now recognised as a pollutant in both national and international legislation (e.g., US National Environment Policy Act and European Commission Marine Strategy Framework Directive). While there is increasing evidence that anthropogenic noise can affect the behaviour and physiology of a wide range of organisms (Barber, Crooks & Fristrup, 2010; Kight & Swaddle, 2011; Morley, Jones & Radford, 2014; Slabbekoorn et al., 2010), it is often difficult to translate the obtained findings into ultimate fitness consequences (Morley, Jones & Radford, 2014). What is needed to move forward are studies that use carefully controlled experiments to rule out potential confounding factors, involving repeated or chronic exposure since cumulative effects may alter responses (Bejder et al., 2009), and that directly assess reproductive success or survival (Francis & Barber, 2013; Morley, Jones & Radford, 2014).

Here we investigate how chronic playback of additional noise, from original recordings of motor boats, affects hatching success and the growth and survival of young in a model fish species Neolamprologus pulcher. Organisms are generally well adapted to tolerate normal environmental fluctuations and challenges early in life (Gilbert, 2001; Hamdoun & Epel, 2007), but anthropogenic disturbances can push conditions beyond usual variability. Heavy metals, extreme temperature or pH, and chemical pollutants have all shown to have a negative impact on development (Baradaran-Heravi et al., 2012; Hamlin & Guillette Jr, 2010; Markey et al., 2001). Noise too can be deleterious during development in humans (reviewed in Ising & Kruppa, 2004) and rats (reviewed in Kight & Swaddle, 2011). However, experimental investigations of how anthropogenic noise impacts early-life in other organisms are rare (but see Banner & Hyatt, 1973; Caiger, Montgomery & Radford, 2012; McDonald et al., 2014; Nedelec et al., 2014; Wysocki et al., 2007).

All fishes studied to date are capable of hearing, with many hundreds known to use and produce sounds (Popper & Fay, 2011; Slabbekoorn et al., 2010), and there is mounting evidence that at least some species are vulnerable to the impacts of anthropogenic noise (Radford, Kerridge & Simpson, 2014; Simpson, Purser & Radford, 2014; Slabbekoorn et al., 2010). Small boats are ubiquitous wherever humans live near aquatic environments, and coastal regions are experiencing unprecedented human population growth: densities within 100 km of the ocean are now three times greater than the global average (Small & Nicholls, 2003). Moreover, aquatic recreation and tourism activities are rapidly rising, with motor boats accounting for a large percentage of boating traffic (see Whitfield & Becker, 2014). Recent studies have demonstrated that motor boat noise can affect the behaviour and physiology of various fish species (Bruintjes & Radford, 2013; Graham & Cooke, 2008; Holles et al., 2013; Picciulin et al., 2010; Radford et al., in press; Voellmy et al., 2014a).

Neolamprologus pulcher is a group-living fish found all around the shores of Lake Tanganyika, East Africa (Duftner et al., 2007) in depths ranging from 3 to 45 m (Taborsky & Limberger, 1981), including harbours and other areas with intensive boat traffic. Previous work has shown that adult behaviour of this species is affected by playback of motor boat noise (Bruintjes & Radford, 2013). Combined with their wide use as a model study organism, including numerous field and laboratory studies (e.g., Bruintjes et al., 2011; Bruintjes, Hekman & Taborsky, 2010; Bruintjes & Taborsky, 2008; Bruintjes & Taborsky, 2011; Heg, 2008; Zoettl et al., 2013; for a review see Wong & Balshine, 2011), N. pulcher provides an ideal opportunity to conduct controlled experimentation at early life stages. Here, in a laboratory experiment, we split the same clutches between two different sound treatments (playback of recordings of motor boat noise or playback of no noise) and predicted that additional noise would reduce hatching success and growth, as well as increase post-hatching mortality, compared with a quieter control condition.

Methods

Ethics statement

The research carried out followed the Association for the Study of Animal Behaviour Guidelines for the Use of Animals in Research. The University of Bristol Ethical Committee (University Investigator Number: UB/10/034) approved all procedures and all effort was made to minimize suffering.

Study animals and husbandry

Groups of three N. pulcher (a dominant pair and one subordinate) were housed in separate 70 l aquaria (size: 71 × 38 × 30 cm) at the University of Bristol, UK and allowed to breed (see Bruintjes & Radford, 2013). To minimize noise levels, the aquaria were placed on 9 mm thick insulation material (Acoustalay 250), external water filters were used (Eheim Ecco 2032) with their inlets placed underneath the water surface, and all aquaria bottoms were covered with a layer of sand (3 cm, 1 mm grain size). The aquaria contained two flower-pot halves (diameter 10 cm) that served as breeding substrate and shelter, and an opaque partition behind which a water heater was placed (Rena smart heater, 100 W). All fish were fed with TetraMin flake food (five times/week), frozen bloodworms (once/week; www.ccmoore.com) and ZM-300 food (once/week; zmsystems.co.uk). Water temperature was kept at 27 ± 0.1 °C and water quality was kept constant. The presence of eggs was checked every 1–2 days in the morning (at 27 °C, N. pulcher eggs take about three days to hatch; Taborsky, Skubic & Bruintjes, 2007).

Experimental design

Eleven groups produced 20 clutches (1–3 per group) during the three month period of the experiment. One day after laying, eggs were counted and the clutch was randomly assigned to receive four weeks of sound treatment (see Playback Files) starting at one of two times. Playback started either (a) immediately, and thus during both the egg- and fry stages (n = 10), or (b) two days post-hatching, and thus during the fry stage only (n = 10). Half of each clutch was assigned to one of two breeding containers, one for each sound treatment, and reared in standard conditions without parental care. The breeding containers (size: 13.5 × 13.5 × 14 cm) were made of thin plastic (0.3 mm) with a fine mesh at the bottom (mesh size 0.5 mm) to ensure aeration and provided ample space for all eggs and fry. Each container was placed in a separate aquarium, with four aquaria used for the control treatment (no additional noise) and four for the additional-noise treatment.

Each aquarium was fitted with an Aqua30, DNH underwater speaker playing either no sound (control treatment) or five randomly chosen 1 h files of noise derived from original recordings of motor boats (additional-noise treatment) at random hours each day during the 13 h light period of the light:dark cycle (see Fig. 1 for spectral level densities and Table 1 for details of the recordings). This intermittent additional-noise regime was chosen because (1) current evidence shows that non-predictable stressors, such as noise, have a stronger effect than predictable (continuous) stressors (e.g., Wright et al., 2007), and (2) anthropogenic noise is typically sporadic in Lake Tanganyika (R Bruintjes, pers. obs., 2005, 2006). All 70 l aquaria (same dimensions as above) for rearing the eggs and fry had similar water quality and water temperature. Eggs and fry were checked daily to establish hatching success and to remove dead eggs and fry; all eggs were removed three days after the first egg had hatched. Fry were fed ad libitum with fry food (ZM-300, see before).

Figure 1 Spectral densities of field and tank-based recordings.

Spectral level densities in an experimental aquarium during playback of an additional-noise track (AN aquarium) and playback of no noise as an ambient control condition (Ambient aquarium), as well as the spectral level densities from recordings made in Lake Tanganyika during the passing of a boat (BN lake) and during an ambient condition without additional boat noise (Ambient lake). The spectral level densities were created using Avisoft Saslab pro (FFT analysis: spectral level units, Hann evaluation window, 50% overlap, FFT size 1024, averaged from a 15 s sample of each recording, presented are 43 Hz intervals).

Table 1 Boat sizes and engine types.

Data on boat sizes and engine types recorded in Bristol harbour (United Kingdom) and Mpulungu harbour in Lake Tanganyika (Zambia). All passing boats were recorded while cruising at average speed 10–50 m from the hydrophone.

Boat number	Place	Boat size (m)	Engine	
1	Bristol harbour	2.0	Outboard, 50 hp, Yamaha	
2	Bristol harbour	3.0	Outboard, 25 hp, Mariner	
3	Bristol harbour	4.5	Outboard, 50 hp, Yamaha	
4	Bristol harbour	8.0	Inboard, 40 hp, unknown brand	
5	Bristol harbour	11.0	Inboard, 40 hp, unknown brand	
6	Bristol harbour	12.0	Inboard, 120 hp, unknown brand	
7	Bristol harbour	12.0	Outboard, 25 hp, Mercury	
8	Bristol harbour	12.0	Inboard, 70 hp, unknown brand	
9	Bristol harbour	14.0	Inboard, 70 hp, Ford Fiesta	
10	Bristol harbour	14.0	Inboard, 50 hp, Ford fsd marine diesel	
11	Bristol harbour	14.0	Inboard, 50 hp, unknown brand	
12	Bristol harbour	15.0	Inboard, 41 hp, Mitsubishi diesel	
13	Bristol harbour	15.0	Inboard, 40 hp, unknown brand	
14	Bristol harbour	18.0	Inboard, 60 hp, unknown brand	
15	Bristol harbour	23.0	Inboard, 80 hp, unknown brand	
16	Bristol harbour	28.0	Inboard, 75 hp, unknown brand	
17	Bristol harbour	28.0	Inboard, 80 hp, unknown brand	
18	Mpulungu harbour	3.5	Outboard, 25 hp, Mercury	
19	Mpulungu harbour	20.0	Inboard, 40 hp, unknown brand	
20	Mpulungu harbour	20.0	Inboard, 60 hp, unknown brand	
21	Mpulungu harbour	26.0	Inboard, 40 hp, unknown brand	

Hatching success was established for the cohort of 10 clutches that were exposed to the control and additional-noise playback during the egg stage; fry survival was determined at the end of sound treatment for all clutches. Surviving fry were photographed with a known size reference to determine total length using tpsDig 2.16 software. They were then sacrificed and dried for 36 h at 70 °C on a Petri dish before weighing them to the nearest 0.0001 g with a precision scale (Mettler AE260, DeltaRange). Between weight measurements, the fry were transferred to a sterile Petri dish to exclude the possibility that debris present in the tank water might have biased our results. Between-measurement error of the weights using the original and new sterile Petri dish was very small (±0.0088%) and the mean of the two measurements was used for further analysis. Mean dry weight was calculated by dividing the total dry weight of the surviving fry per clutch per treatment by the number of fry. Mean fry length was calculated by averaging the length of the surviving fry per clutch per treatment.

Playback files

Original recordings of motor boats were made in Bristol harbour (UK) during the passing of boats of similar size and with similar engines as found in the harbour of Mpulungu, Lake Tanganyika, Zambia (Table 1); see also Bruintjes & Radford (2013). Recordings were made with an omnidirectional hydrophone (HiTech HTI 96-MIN with inbuilt preamplifier; manufacturer calibrated sensitivity −164.3 dB re 1 V/µPa; frequency range 2–30,000 Hz) and a recorder (Roland Edirol R09HR; 24-bit; sampling rate 44.1 kHz; calibrated using a single reference of known amplitude). All acoustical analyses were done with Avisoft-SASLab Pro software version 5.1.17 (Avisoft Bioacoustics, Berlin, Germany). Ten different 15 min sound files were created using three randomly chosen boat passes from a pool of 17 recordings (mean ± SE duration of boat passing = 18 ± 3 s, using two boat passes per minute). Following this, the 15 min files were used to create one-hour playback tracks, five of which were played back in the additional-noise treatment aquaria per day. The recordings were lowpass filtered at 2 kHz to minimize resonant frequencies (Batty, 1989), and highpass filtered at 100 Hz to play within the frequency range of the speaker (see below).

The tracks were played back in the experimental aquaria using a laptop computer with an external soundcard (Roland Edirol UA-1EX), an underwater speaker (Aqua30; DNH, effective frequency range 80–20,000 Hz) and were re-recorded in the centre of the aquarium. The sound levels of the individual recordings were adjusted to create files of approximately equal spectral level densities and the recorded tracks were modified so that they were within 5 dB re 1 µPa root mean square (RMS) of one another. The files were adjusted to play at 127 dB re 1 µPa RMS (mean ± S.E.: 127.2 ± 0.5 dB re 1 µPa), calculated over the loudest 2 s per playback. An example of the spectral level densities of the recordings in the field and in the aquaria is given in Fig. 1.

Statistics

Statistical analyses were performed on all 20 clutches and on the first clutches per pair (n = 11) with PSAW 18.0.0, using proportions for hatching success and survival, and means per clutch for length and weight (see Table 2 for individual sample sizes; length data are missing for one clutch). Linear Mixed Models (LMMs) with REML variance component estimation were used to control for repeated measures as multiple clutches were laid by most groups. In all analyses, ‘sound treatment’ (additional-noise or control) was considered as a fixed factor and for fry survival, length and weight, we also controlled for fixed factor ‘exposure start time’ (during the egg or fry stage), since those exposed only post-hatching were older at assessment; the interaction term between sound treatment and exposure start time was never found to be significant, and not reported in the Results. As random factors, where appropriate, clutch number, group and clutch number nested within group were included.

Table 2 Sample sizes.

Individual sample sizes used for analyses of hatching success, survival, and length and weight after four weeks of exposure to additional-noise and control conditions.

Descriptive data	Hatching
success	Fry survival
at week 4	Fry measurements at week 4
(length & weight)	
No. of clutches used	10	20a	20a	
Total no. eggs or fry	109 fry hatched
out of 237 eggs	191 fry survived
out of 415 eggs	191 fry survived
out of 415 eggsb	
Range of egg or fry
no. per clutch	7–34 eggs;
0-34 fry	7–34 eggs;
0-30 fry	7–34 eggs;
0-30 fry	
Notes.

a In 19 out of 20 clutches at least one fry survived in one or both treatment(s).

b Length of one clutch was not taken, resulting in 179 length measurements.

Results

Hatching success was not significantly affected by sound treatment (LMM: F1,9 = 0.44, p = 0.838; Fig. 2A). After controlling for exposure start time (F1,18 = 8.13, p = 0.011), fry survival was not significantly affected by sound treatment when considering all 20 clutches (F1,18 = 0.13, p = 0.724; lower fry survival in the additional-noise treatment in eight cases; lower survival in the control treatment in eight cases; no difference between treatments in four cases). Qualitatively similar results were found when considering only the first clutches per pair (exposure start time: F1,9 = 8.94, p = 0.015; sound treatment: F1,9 = 0.31, p = 0.590; Fig. 2B).

Figure 2 Hatching success, survival, length and weight at four weeks.

Proportion of clutch that successfully hatched (n = 10) (A), proportion of clutch that survived to four weeks (B), mean length of fry at four weeks (C), and mean dry weight of fry at four weeks (D) in the two sound treatments. Presented in (B)–(D) are values from first clutches (n = 11), with solid lines representing clutches receiving noise starting during the fry stage and dotted lines clutches receiving noise starting during the egg stage. Four clutches in (A) and one in (B) had no hatching success or survival, but are shown above zero for visualization. In (C) and (D), unconnected ‘x’ symbols represent fry surviving in one of the treatments.

After controlling for exposure start time (LMM, length: F1,28.9 = 10.74, p = 0.003; weight: F1,30.7 = 16.2, p < 0.001), sound treatment did not significantly affect either fry length (F1,20.5 = 0.78, p = 0.388) or weight (F1,22.3 = 1.08, p = 0.661) when considering all 20 clutches. Fry in the additional-noise treatment were lighter in six cases and heavier in 10 cases than in the control treatment, while they were shorter in eight cases and longer in seven cases in the additional-noise treatment compared to the control treatment; in the remaining four clutches, no fry survived in at least one of the treatments. When considering only the first clutches per pair, qualitatively similar results were found, with sound treatment not significantly affecting either fry length (F1,13 = 0.54, p = 0.475; exposure start time: F1,13 = 0.26, p = 0.616; Fig. 2C) or weight (F1,13 = 0.13, p = 0.728; exposure start time: F1,13 = 0.49, p = 0.497; Fig. 2D).

Discussion

We found no evidence in our laboratory study using Neolamprologus pulcher that hatching success or fry survival and size four weeks post-hatching were detrimentally affected by chronic exposure to playback of additional noise originating from recordings of small motor boats. These findings are in line with those of Wysocki et al. (2007), who did not detect any significant impact of high continuous aquaculture noise (arising from filters, aeration and water pumps) on the growth or survival of rainbow trout (Oncorhynchus mykiss), but contrast the work of Banner & Hyatt (1973), who reported decreased hatching success and larval growth in fish reared in tanks with high continuous sound-pressure levels from water-pump noise. Our work adds to these previous fish studies by utilising a split-brood design, thus ruling out potential genetic confounding effects, and by considering a sporadic noise source, which may potentially have a greater impact than continuous noise sources of the same intensity (Francis & Barber, 2013; Wright et al., 2007).

Using the same sound playbacks, intensity levels and aquaria as in the current study, Bruintjes & Radford (2013) found that additional noise significantly affected the behaviour of N. pulcher adults: decreased nest-digging and anti-predator defence was observed, and there were impacts on intra-group aggression levels. One possibility for the apparent lack of response to noise exposure during early life may therefore be that the embryos and fry of this species have yet to develop the hearing capacity to detect the experimental sounds. However, while measurements of N. pulcher hearing thresholds at different developmental stages are not available, fry of several fish species are attracted to reef noise (Simpson et al., 2005a) and embryonic coral reef fish respond to noise (Simpson et al., 2005b). It is also unlikely that the early developmental stages are more robust than adults to anthropogenic disturbances (Etzel et al., 1997; Moller & Swaddle, 1998). The provision of ad lib food might have buffered the potential effects of noise; reduced food finding or handling efficiency (Purser & Radford, 2011; Voellmy et al., 2014b) might create negative fitness consequences in natural conditions. It is also possible that there were undetected effects of the additional noise. For instance, the growth trajectories of the fry in the two sound treatments might have differed, especially if responses to noise change with time (Wale, Simpson & Radford, 2013). Any compensation for initial slow growth could result in consequences for individual fitness (Metcalfe & Monaghan, 2001), but this requires future investigation.

Our experiment was performed in aquaria, making it possible to control carefully various potential confounding factors (Slabbekoorn, in press). However, the acoustics of small spaces are complex and are dominated by the particle velocity element of the sound field (e.g., Parvulescu, 1964). While that might mean that stronger effects would be expected than in natural conditions, field-based studies in the far field and using real noise sources are vital to assess fully the potential impact of anthropogenic noise, especially from the perspective of policy making and management. For now, our results suggest that chronic noise exposure does not necessarily have direct negative impacts on early-life survival and growth. Studies of the effect on individual fitness are crucial in their own right, as well as forming the basis for assessments about population viability and resilience in the face of anthropogenic change. Given the mixed results from studies investigation the potential impact of noise on early life and development (e.g., Banner & Hyatt, 1973; Nedelec et al., 2014; Wysocki et al., 2007; this study), we advocate further detailed studies in the near future.

Supplemental Information

Supplemental Information 1 Raw data

Click here for additional data file.

We are grateful to Markus Zöttl and Michael Taborsky for help importing the fish, Martin Genner for the breeding containers, and Steve Simpson, Julia Purser, Irene Voellmy and Sophie Nedelec for helpful discussions and comments.

Additional Information and Declarations

Competing Interests

Author Contributions

Animal Ethics

The authors declare there are no competing interests.

Rick Bruintjes conceived and designed the experiments, performed the experiments, analyzed the data, contributed reagents/materials/analysis tools, wrote the paper, prepared figures and/or tables, reviewed drafts of the paper.

Andrew N. Radford conceived and designed the experiments, contributed reagents/materials/analysis tools, wrote the paper, reviewed drafts of the paper.

The following information was supplied relating to ethical approvals (i.e., approving body and any reference numbers):

University of Bristol Ethical Committee Investigator Number: UB/10/034.

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
