# Peer review of "Chronic playback of boat noise does not impact hatching success or post-hatching larval growth and survival in a cichlid fish"

_PeerJ, doi:10.7717/peerj.594_

## Round 0.1 · original submission · Minor Revisions

All three reviewers found the manuscript interesting and relevant. Each reviewer provides useful suggestions, mostly to provide more detail and improve clarity of methods and findings.

Reviewer 1 ·

Basic reporting

Overall, I found this manuscript to be interesting and mostly very well written. The study will be of interest to many researchers in the field of effects of anthropogenic noise on aquatic animals, and the role of stressors in early-life stages.

The author may want to include a couple of studies that I feel is relevant literature:

Line 33 other experimental investigations of how anthropogenic noise impacts early-life stages....could include Caiger et al. 2012 - Chronic low-intensity noise exposure affects the hearing thresholds of juvenile snapper.

And potentially something from the invertebrate literature?
E.g., McDonald et al. 2014 - Vessel generator noise as a settlement cue for marine biofouling species

Experimental design

Great replication on a number of different boats used during playback!!

Potential confounding factors in experimental design were noted by the authors and minimized where possible. However, I have a question relating clarity in the methods for tank replication:

Starting line 82 & 125: It is not 100% clear in the text how many additional aquaria are used for the boat noise treatments in the experiment. It states a total of 8 are used so is that 4 each? Please make this more clear in text as currently it is possible to interpret that only one was used, which I don't think is the case as it is far from ideal as there is potential for the one speaker or tank spreading a potential artifact to all noise treatments.

Validity of the findings

The aim of the proposed study was to investigate the effects of additional boat noise on the hatching success, post-hatching mortality and growth on the early life stages of Neolamprologus pulcher in a laboratory experiment, which I believe it had achieved.

I also agree with the author when saying, the use of field-based studies, using actual noise sources are vital to fully assess the potential impact of anthropogenic noise on a species, especially when the results are to be used to assist policy making and management. However, this study still gives a robust measure of the effects of noise to this species in a laboratory setting, and adds it the literature on this topic, especially seeing it suggests the noise exposure doesn't necessarily have direct negative impacts on the early-life histories of some species.

Reviewer 2 ·

Basic reporting

Overall I thought this manuscript was very interesting and the lack of effect of sound on N. pulcher hatching success of larval growth actually quite surprising.
I found the extensive use of parentheses made the manuscript hard to read and disrupted the flow.
Also, the authors need to remove the excessive use of 'We'
Figures were relevant
Table 1 presented a lot of information about boat size and outboard engine type, however there was no connection between that data and the actual sound emitted by each engine. The results observed may be significantly different with different boat engines as they would each produce different sound signatures. Research has indicated that vessel sound within the 100-1000 Hz frequency range does induce behavioural changes on larvae, however its not clear which outboard engine produces sound in that range. This really needs to be clarified.

Experimental design

Overall I thought the experimental design was ok. However I was concerned at what the authors defined as 'ambient noise', as this seemed to be quite noisy. Where did the background ambient noise in the aquarium come from?

Validity of the findings

I agree with the authors, further work is required to understand the effects of noise on the larvae.

Annotated reviews are not available for download in order to protect the identity of reviewers who chose to remain anonymous.

Reviewer 3 ·

Basic reporting

Figure 1 resolution should be improved, or the "Ambient aquarium" line modified to be more visible. The figure caption parentheticals and legend also do not match up: AN aquarium does not appear in the figure legend but is in the caption. I believe that the "BN aquarium" in the figure legend should actually read "AN Aquarium".
Additionally, more information is needed in the figure caption regarding the recordings and method used for power spectra generation to properly interpret them, especially given how smoothed they appear (surprising to see such low variability, particularly in a tank, so I would like to better understand how the spectra were produced). What is the analysis window length and how long were the total recording samples used for this analysis (i.e. how many windows, of what length, were used to generate what I assume are mean spectra)? Line 120 indicates that the treatment track was a random 1-hr compilation (from four different 15-min recordings? please clarify), so it is relevant to know whether the example spectra shown is for a full hour or a 15-min sound file or less.

Line 129 - This description of the tank treatment calibrations should be expanded to clarify the intent and interpretation. Were the replayed recordings adjusted to match the original recording SPL or just one another (wording is ambiguous)? The accompanying figure implies that the tank treatment was set to match the original recording as possible, but the text is not clear on this. What was the target level for all playbacks, and how was it determined if presumably the 10 tracks varied somewhat in their original levels and spectra?
Line 130 - I assume you modified the full bandwidth sound levels so that they were within 5 dB (as opposed to all parts of the spectra, which would differ between recordings of different boats). In this case, the units would be dB re 1µPa rather than per Hz.

Experimental design

There are several aspects of the experimental design that are unclear from the description in the methods, and they should be straightforward to clarify in the text:

Line 81: I assume that half of each clutch was then also assigned to each treatment? Or half-clutches from the same group to the (a) and (b) designs? This aspect is not communicated.
Line 93: Not sure how twenty clutches were divided into eight experimental aquaria based on design outlined, and since each clutch was divided into two would that mean there were ten replicate clutches?
Line 105-106: which two measurements? Is this for the clutch divided into two?

Validity of the findings

No Comments.

---

## Round 0.2 · accepted · Accept

Your edits have been thorough and responsive to reviewer comments.